# The impacts of COVID-19 pandemic on public transit demand in the United States

Luyu Liu[1,2]*, Harvey J. Miller[1,2], Jonathan Scheff[3]

**1** Department of Geography, The Ohio State University, Columbus, Ohio, United States of America, **2** Center for Urban and Regional Analysis, The Ohio State University, Columbus, Ohio, United States of America, **3** Transit App Inc., Montreal, Quebec, Canada

* liu.6544@osu.edu

**Data Availability Statement:** The daily transit demand data, COVID19 confirmed case data, census demographic data, Google search trend data, values behind the reported average and graphs, and code are available at http://dx.doi.org/10.34740/kaggle/ds/687971. The hourly transit

## Abstract

The COVID-19 pandemic and related restrictions led to major transit demand decline for many public transit systems in the United States. This paper is a systematic analysis of the dynamics and dimensions of this unprecedented decline. Using transit demand data derived from a widely used transit navigation app, we fit logistic functions to model the decline in daily demand and derive key parameters: *base value*, the apparent minimal level of demand and *cliff* and *base points*, representing the initial date when transit demand decline began and the final date when the decline rate attenuated. Regression analyses reveal that communities with higher proportions of essential workers, vulnerable populations (African American, Hispanic, Female, and people over 45 years old), and more coronavirus Google searches tend to maintain higher levels of minimal demand during COVID-19. Approximately half of the agencies experienced their decline before the local spread of COVID-19 likely began; most of these are in the US Midwest. Almost no transit systems finished their decline periods before local community spread. We also compare hourly demand profiles for each system before and during COVID-19 using ordinary Procrustes distance analysis. The results show substantial departures from typical weekday hourly demand profiles. Our results provide insights into public transit as an essential service during a pandemic.

## Introduction

COVID-19, a novel coronavirus disease, emerged in December 2019 to become a global pandemic due to its contagiousness, fatality rates, and lack of effective vaccines or treatments. To deal with the pandemic, from early February 2020 the U.S. Centers for Disease Control and Prevention (CDC) recommended social distancing, self-quarantine, and working from home to stop the spread of the virus. States and cities followed these and other guidelines, closing schools and businesses, and issuing calls to stay at home. These sudden and unprecedented shutdowns led to declines in travel demand at all geographic scales and across all modes [1, 2].

Public transit is particularly vulnerable to disruption and shocks from pandemics due to the collective nature of its mobility. News articles and preliminary reports from transit agencies suggest this is the case with COVID-19. For example, in Washington DC, Metrorail

demand data is proprietary. The archive includes instructions for requesting these data from the third party.

**Funding:** [Funding Statement]: The authors received no specific funding for this work. The third author Jonathan Scheff, who helped to edit and review the draft and provide hourly transit demand data, is currently employed at Transit app Inc. The funder, Transit app Inc., provided support in the form of salaries for the third author Jonathan Scheff, but did not have any additional role in the study design, data collection and analysis, decision to publish, or preparation of the manuscript. The specific roles of the third author are articulated in the 'author contributions' section, which only include "data curation" and "writing – review and editing". We confirm that the commercial affiliation of the third author (Transit app Inc.) did not play a role in our study and there has been no significant financial support for this work that could have influence its outcome.

**Competing interests:** [Competing Interests Statement]: The first two authors, Luyu Liu and Harvey J. Miller, declare no competing interests. The third author Jonathan Scheff is currently employed at Transit app Inc. as a data scientist. We confirm that this commercial affiliation does not alter our adherence to all PLOS ONE policies on sharing data and materials.

ridership declined by 90% and bus ridership declined by 75% by the end of March 2020 [3]. In contrast, some transit agencies experienced modest declines in ridership. For example, the ridership of VIA Metropolitan Transit in San Antonio, Texas only declined by 30% by the end of March, 2020 [4]. These disparate declines reflect varying degrees of public transit dependence across communities. They also suggest different vulnerabilities of transit systems to shocks since drops in fare box revenue may lead to subsequent cuts in services, particularly since cash-strapped local governments may not have the ability to continue their support.

Declines in transit demand are also unequal across social groups, since many information, managerial, tech, and knowledge workers can work from home while people with jobs that demand physical presence still need to travel to work [5]. The remaining public transit users are likely transit dependent riders who require public transit for mobility and accessibility to jobs, health care, and services [6]. Since only essential businesses and services were open during this period, these dependent riders are also likely performing necessary activities for themselves and society, highlighting the nature of public transit as a critical service [7]. These dependent riders traveling to perform essential jobs may also have a different hourly demand profile than the demand profile experienced by transit agencies during normal times, reflecting a potential mismatch between their needs and transit services [8].

The differential impacts of pandemics on public transit demand is an underexplored question. There is limited research based on the experience of Asian cities during recent pandemics. During the 2003 SARS pandemic, the Taipei underground system lost almost 50% of daily ridership during the peak of the pandemic [9]. An analysis of Seoul transit system smart card transaction data during the 2015 MERS outbreak shows variations in the decline in trip frequencies across different public transit modes, social groups, and neighborhoods [10]. However, the literature is still scarce: there are no studies systematically investigating a pandemic impact on transit demand across communities on a national scale.

COVID-19 provides an unfortunate but imperative juncture to understand the differential impacts of a pandemic on public transit demand across communities and social groups. In the past, this was difficult since many transit authorities do not publish or otherwise make ridership data readily available [11]. Ridership data may be available from some agencies by request; however, these data are often defined and measured in different ways [11, 12], making comparisons across agencies problematic. This data constraint has been eased by the rise of third-party transit navigation applications, enabled by transit agencies publishing their schedules and real-time vehicle information. Transit demand data in the form of queries to a transit navigation app used across multiple communities provide a consistent benchmark to make comparisons.

In this study, we use the data from the Transit app, a popular mobile phone-based transit planning app, to conduct a systematic analysis of the impacts of COVID-19 on 113 public transit systems across the United States. We fit logistic curves to describe the decline in daily transit demand across public transit systems, extracting key parameters: i) *base value*, the apparent minimal level of demand; ii) *cliff and base point*, representing the initial date when decline in transit demand began and the final date when decline attenuated, respectively; and iii) *decay rate*, representing the speed of the demand decline. We use regression and correlation analyses to relate the base values to socioeconomic and demographic factors in each community. We also compare the distance between the cliff/base points and the first date of local community spread to measure whether public transit users in different metro areas reacted at different speeds to the unfolding pandemic. Finally, we use hourly transit demand data to capture COVID-19's impact on daily patterns of transit demand; we measure the similarity of hourly demand profiles during the COVID-19 pandemic relative to the adjusted normal demand profiles. We conclude that COVID-19 had substantial but uneven impacts on transit systems and

social groups that can be explained via a social equity lens. Based on this analysis, we propose future directions for understanding the impacts of pandemics on public transit.

## Data and methods

In this section, we describe the primary data sources in our study, namely, Transit app demand data and COVID-19 case numbers. We also describe our model of daily transit demand decline, the logistic curve. From these fitted curves, we derive several parameters describing the declines in daily transit demands: i) *base values* measuring the apparent minimum levels of transit demand; ii) *cliff and base points* indicating when demand decline started and stopped; iii) the *decay rate* measuring the speed of transit demand decline, and; iv) *response intervals* capturing the time lags between the first reported case in a community with respect to when decline started and stopped. Moreover, we describe *ordinary Procrustes analysis* for measuring differences in hourly travel demand during and before the COVID-19 pandemic, and weekday versus weekend demand during the pandemic.

### Data sources

**Transit demand.**   Since it is difficult to obtain comprehensive public transit ridership data at a national-scale, we use data from the Transit mobile phone app (transitapp.com) as an indicator of changes in daily and hourly transit demand. Transit is a popular mobile phone app providing real-time public transit data and trip planning. The app covers over 200 cities around the world and more than 60 US metro areas (discussed below) with more than five million downloads on Android platform [13] and 73.5 thousand ratings on Apple App store [14]. We accessed daily demand data from Transit via their daily updated webpage: these are change values expressed as a set of percentage of app usage relative to the same date last year, adjusted for annual growth [15]. These estimates use app activity as a proxy for transit ridership demand.

Transit estimates their market share of transit users to be an average of 8% in the United States. To assess Transit app usage data as a measure of transit demand, we compared ridership decrease reports derived from individual transit systems' websites and local news outlets. Most transit systems do not release estimates daily; instead, many report single estimates for a given date. We compared these ridership decrease reports with the corresponding estimates from the Transit app data on the same dates for 40 transit systems with published ridership reports. The average difference between the Transit app estimate and agency reported value is 3.7%; a paired T-test indicates that we cannot reject the null hypothesis that the mean difference is zero ($p = 0.14 > 0.05$). However, the standard deviation of the differences is 15.96%; this may be due to the varying definitions of normal ridership level among agencies. Although the test suggests the Transit app data are a good approximation of public transit demand, it is important to note that both the Transit app data and transit agencies' data are inferences. Ridership counts from agencies, especially daily ridership counts (as opposed to monthly) vary in methodology. Moreover, agency counts during COVID-19 are particularly prone to error due to back-door only boarding, fare-free service, and other pandemic-related changes that affect the accuracy of ridership counts. Transit app activity does not include individuals who cannot afford a smart phone and data plan, cannot use the app due to different abilities, or choose not to use it. It is likely that our measures are underestimates of transit demand decline, especially among more disadvantaged social groups and older populations. Compensating for these disadvantages is the large Transit app user base across most transit systems in the United States, allowing comparison across transit systems using a common benchmark.

The daily Transit data include demand decrease estimates for 182 public transit systems across the United States, Canada, Australia, New Zealand, and France. We selected 113 county-level transit systems in 63 metro areas and 28 states across the United States. We excluded 7 state-level or cross-county systems if their ridership could draw from large and geographically diverse areas, such as Pacific Surfliner, which extends to the entire South California coast, and Metro-North Railroad, which crosses multiple counties and states in the Northeast United States. The time period of daily data is from February 15th to May 17th. We also use hourly demand decrease estimates for 93 public transit systems across the United States. The time frame for the hourly data is from March 16th to May 10th. None of the data collected or accessed in this study in this study contain individual personal information.

**COVID-19 case numbers.** We use the daily case numbers for each county from COVID-19 maps and county-level dataset produced by USAFacts [16]. The data include all county-equivalents' confirmed cases in the US on a daily basis. To find the linkage between the case numbers and the demand decrease, we geocoded each transit system to its corresponding county-equivalent.

## Logistic function for daily transit demand change

Fig 1 shows the varying temporal pattern of all US systems' average transit demand from Feb 15th to May 10th. We see a pattern of stable demand before the COVID-19 crisis, a period of decline, followed by re-stabilization at a lower demand level. This is a pattern described well as a logistic (anti-) growth process, expressed using a logistic or sigmoid function:

$$f(x) = \frac{B}{1 + e^{-k(t-t_0)}} + b \tag{1}$$

where $b$ is the pre-COVID stable demand level, $B$ is the re-stabilized demand level after the

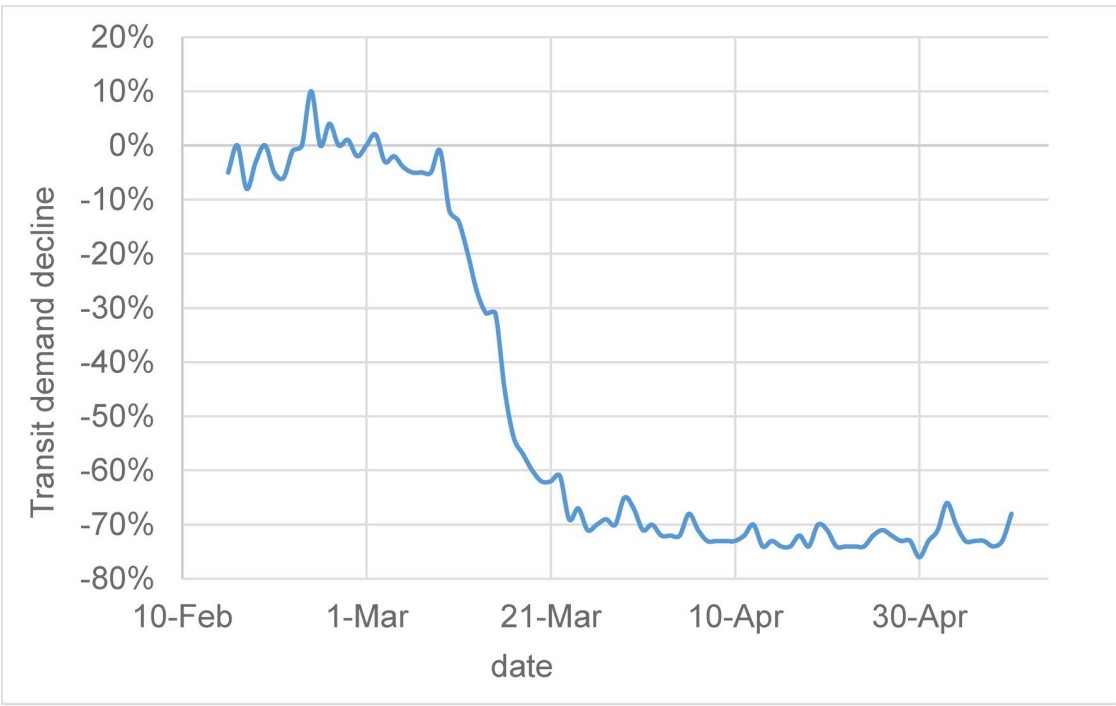

**Fig 1. Temporal pattern of average transit demand during the study period.**

decline period, $k$ is the decline rate; $t$ is time (day) and $t_0$ is the time when the function reaches the midpoint. We fit each transit system's demand data to the logistic function using a least square optimizer individually. We tested the goodness of fit using the R-squared between actual demand and fit values and Q-Q plot.

Fig 2 shows an example of a fitted curve and illustrates the key parameters we extract from these curves. In the following sections, we describe the semantics of these parameters within our application context.

**Base value.** The parameter $B$ represents the relative decrease from the normal level as demand re-stabilizes at a lower level; we define it as *base value* as visualized in Fig 2. This represents the ratio of public transit users in the system who still use the transit system regardless of the pandemic. This demand level is not necessarily a persistent state: demand may destabilize and grow again due to external factors, such as re-opening of businesses or stay-at-home fatigue. The base value represents the base level of demand from the initial shock to the system.

We examine relationships between the estimated base values and socioeconomic factors using linear regression analysis. The county-level socioeconomic data are from the latest American Community Survey (ACS) 5-year estimate table (2014–2018). We derived several socio-economic indicators. First is the *ratio of population with non-physical occupations*. Similar to *life fixity* [10], this measures the population's degree of freedom to change the routine of their daily life: it represents how many people can work from home thus avoid regular transit commuting to reduce contagion risk. If a community has higher ratio of non-physical jobs, more workers have the freedom to work from home, meaning that transit demand may

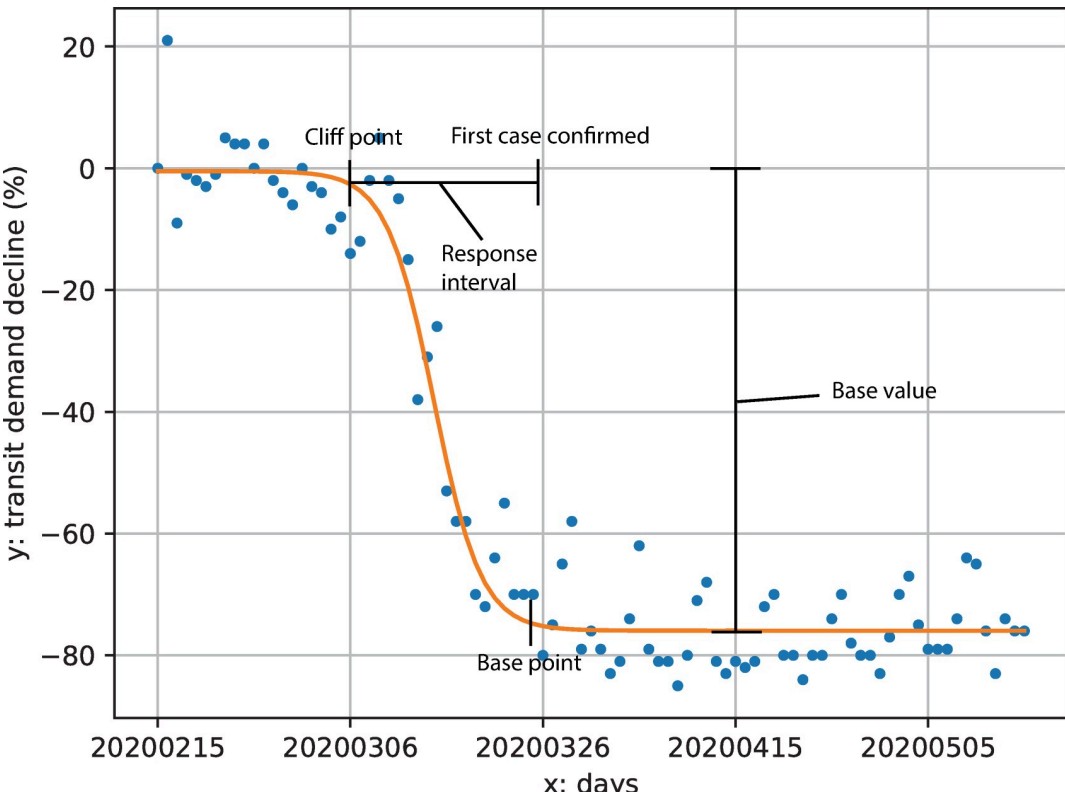

**Fig 2. Fitted curve and key parameters for an example public transit agency (Capital Metro in Austin, Texas).**

decrease more than communities with lower ratios. We use the occupations statistics for employed civilian population 16 years and over from the ACS data. The table contains the number of employed population in different industries. According to the statistics released by US Bureau of Labor Statistics, Information, Financial activities, and professional and business service have the highest proportion who can work from home [5, 17]. Among these occupation categories in the ACS table, we assigned "management, business, and financial operations occupations", "professional and related occupations", and "office and administrative support occupations" as the occupations where people can work from home remotely.

*Income*, *population density*, *and job density are* socioeconomic factors that relate to job composition. Also, transit users tend to skew toward lower incomes in the United States; higher incomes also correlate higher population density and job density at the community scale. We use the median income, population density, and employment density data from ACS.

A third set of indicators is the *ratios of minority and female population*, including African American and Hispanic populations. Many studies demonstrate the disadvantaged status of African American, female, and Hispanic populations in mobility and job accessibility across different metro areas in the United States [18–20]. Therefore, we investigate the relationship between base value and each vulnerable group's ratio. We derive the sex, race, and Hispanic population data from the ACS data.

A fourth socioeconomic indicator is community *age structure*. Older individuals are more at risk of hospitalization and death from COVID-19 [21]. We use 45 years as a threshold to identify high-risk populations. We measure the ratio of people with age over 45 from the ACS 5-year estimates (2014–2018) data.

Moreover, we use measures of *awareness*. If local residents are aware and concerned about COVID-19, the base value may be lower because more people will avoid public transit trips. To test this, we use the Google search trend index to represent the awareness of the local people [22–24]. We collected the average Google search trend data for different designated market area that each transit system locates in for 90 days from January 18[th] to April 17[th] 2020 [25], the latter being the latest day that any system experienced further significant decline. We use "Coronavirus" over "COVID19" as the search keyword for its popularity among the public [26].

Finally, *transit dependency* is also a potential factor affecting the transit system's usage rate [27] and base value. If an area has more people depending on transit, the usage rate of transit during the pandemic is supposedly higher. We derive the ratio of people who transit to work and the percentage of house units with no vehicle access from ACS data to infer transit dependency.

To supplement the analysis, we also refer to the user survey results conducted by Transit app, Inc. about the demography of the passengers during the pandemic [28, 29]. The survey was conducted in early April, 2020 across the United States (n = 15000) and Canada (n = 10000) via the Transit app interface. The survey investigates the age, race (including Spanish speakers), gender, trip purpose, occupation composition of the passengers who remained transit system users during the pandemic [15, 28, 29].

**Cliff/base points and decay rate.** The *cliff point* $t_c$ and *base point* $t_b$ are time points when demand decline started and when it re-stabilized as shown in Fig 2. We calculated these from the fitted logistic curves using confidence interval theory rather than the observed data to provide more stable estimates. We derived these measures by first constructing the probability density function of the normalized logistic function F(x):

$$\forall -\infty < t < \infty, F(t) = \frac{f(t) - b}{B} = \int\limits_{-\infty}^{t} P(t)dx \tag{2}$$

$$P(t > t_b) = 1 - \frac{f(t_b) - b}{B} = \frac{\alpha}{2} \tag{3}$$

$$P(t < t_c) = \frac{f(t_c) - b}{B} = \frac{\alpha}{2} \tag{4}$$

where $P$ is the probability density function of the normalized logistic function; to normalize the logistic function, we subtract the normal value $b$ and divide the result by $B$ to construct the $P$ function so that $\int_{-\infty}^{\infty} P(t) = 1$. $\alpha$ is the confidence threshold. From Eqs (2)-(4), we can see that $P(t_c < t < t_b) = f(t_b) - f(t_c) = 1 - \alpha$; $\alpha = 0.05$ ensures that 95% of the decline falls between the cliff and base points. From the formula, the cliff and base points are:

$$t_c = t_0 - \frac{\ln\left(\frac{2}{\alpha} - 1\right)}{k} \tag{5}$$

$$t_b = t_0 + \frac{\ln\left(\frac{2}{\alpha} - 1\right)}{k} \tag{6}$$

The cliff point is the first day when the demand curve began to diverge from normal—i.e., when transit users start to avoid the transit. The base point is the day when decline slows and transit demand has re-stabilized. We expect the cliff point to be impacted by policies and actions in local communities. We therefore consider the date when each state declared a state of emergency due to COVID-19 and conduct correlation analysis between the emergency declaration date and the cliff point.

Parameter $k$ in Eq (1) represents the rate of transit demand decline; we therefore define it as *decay rate*. It indicates the speed of response from users who have the ability to stay at home or not use public transit. We conduct correlation analysis between decay rate and cliff/base points to examine the relationship between de-stabilization, re-stabilization and the rate of demand decline.

**Response intervals with incubation lags.**   *Response intervals* compare the time of community spread with the initiation (cliff point) and conclusion (base point) of transit demand decline in each system. This measures the responsiveness of transit demand to the pandemic. Although declines in transit demand are not welcome from a revenue perspective, lower demand means fewer people potentially exposed on transit; it also means the remaining dependent riders are less exposed and can practice social distancing more easily. Ideally, a transit system initiates and finishes its demand decline before there is community spread.

Meanwhile, the date of first reported community spread is not necessarily the first date of actual spread due to the incubation period for the disease. The median of incubation period is five days and can be as long as 14 days [30]; the virus can also spread asymptomatically [31–33]. Additionally, the availability of testing kits and response times from local authorities can lead to an underestimation of total cases [34, 35]. Since the incubation period and case load accuracy both vary, we introduce *incubation lag* as a parameter of the response intervals relative to the cliff and base points:

$$r_c(l) = t_s - l - t_c \tag{7}$$

$$r_b(l) = t_s - l - t_b \tag{8}$$

where $t_s$ is the date of first confirmed case in the county of the transit system; $l$ is the incubation

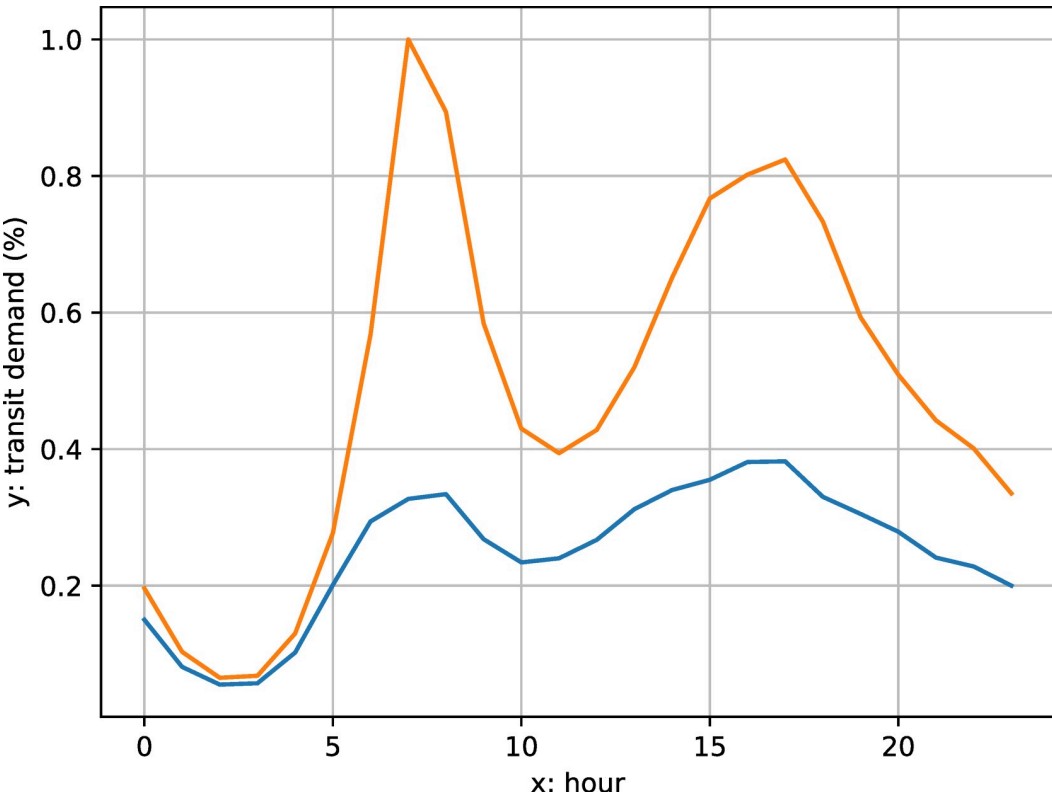

**Fig 3. MTA New York City subway hourly transit demand curves (blue: COVID curve; orange: typical curve).**

lag; $t_c$ and $t_b$ are the cliff and base point; and $r_c$ and $r_b$ are the response intervals relative to the base point and cliff point, respectively. A positive response interval means that the transit users responded earlier than the epidemic spread; the larger the value is, the earlier the transit users responded relative to the onset of the spread of the virus. The cliff response interval indicates transit users' awareness while the base version indicates the response time. For example, a negative base response interval that is large in magnitude indicates that public transit reached its minimum level long after the first confirmed case in the county.

## Change in daily transit demand dynamics

The analyses based on the average daily transit demand shows the coarse-grained temporal variation among different transit systems. Transit demand dynamics within each day can also change during a pandemic. For example, Fig 3 shows changes in transit demand by hour before (orange) and during COVID (blue) for the New York City subway. The higher and peaky curve is a typical US daily travel demand pattern, with morning and afternoon peak demand periods corresponding to commuting to and from work (for the "standard", nine-to-five work day), respectively. In contrast, the COVID demand curve indicates not only lower demand, but less pronounced peak demand periods.

We use the shape analysis technique of *ordinary Procrustes analysis* to measure differences between hourly travel demand during and before the COVID pandemic, and also weekday versus weekend demand during the pandemic. This involves measuring the transformations required to superimpose the two curves. In traditional Procrustes analysis for arbitrary shapes, these transformations are panning, scaling, and rotating [36]. In the case of one-dimensional

curves, only scaling is appropriate. We introduce a stretch factor $p$ as a multiplier to fit two curves so that their Procrustes distance is minimized:

Minimize:

$$S(p) = \sqrt{\sum_{t=1}^{h} (p \cdot f(t) - g(t))^2} \qquad (9)$$

where $S(p)$ is the Procrustes distance between the two curves, $p$ is the stretch factor, $h$ is the number of data points in the dataset, $f(t)$ and $g(t)$ are the two curves' value at time $t$. The solution to this optimization problem is:

$$S(p^*) = \sqrt{\sum_{t=1}^{h} \left( \frac{\sum_{t=1}^{h} f(t) \cdot g(t)}{\sum_{t=1}^{h} f^2(t)} \cdot f(t) - g(t) \right)^2} \qquad (10)$$

The optimal Procrustes distance $S(p^*)$ is a measure of the difference in the shapes of the curves: a larger Procrustes distance means bigger differences in the shape of the two curves.

We measured the Procrustes distances between the normal hourly demand curve and COVID curve for each day between March 16th to May 10th and calculated the average. We also calculated the average Procrustes distance between Wednesdays' (representing a typical weekday) and Sundays' (representing a typical weekend) demand profiles in each week. Under normal travel demand patterns in the US, weekday and weekend hourly demand profiles are different, with no typical twin demand peaks on weekend days. We expect that weekday and weekend public transit demand profiles converged during COVID.

## Results

### Logistic function fit

The R-squared between the actual and fit values shows logistic function's fit accuracy is very high: the median of all R-squared is 0.969 and 110 of 113 systems' R-squared is larger than 0.9. We also use Q-Q plots to test the normality of the residuals: the results show that each system's actual quantiles are close to the theoretical normal distribution quantiles. From these tests, we conclude that the logistic function properly fits the transit demand data.

### Base value

The average base value of 113 transit systems is -72.66% (standard deviation = 11.58%). We express these values as negative differences from previous demand levels: larger negative numbers are lower bases. We can see clear geographic differences in Fig 4: cities in the Deep South and Midwest have higher base values (i.e., negative but smaller in magnitude—less decline in public transit demand). Meanwhile, high tech cities such as the San Francisco Bay area and university cities such as Ithaca, Ann Arbor, and Madison generally have a very low base value (higher decline in public transit demand).

Table 1 provides results from the regression analysis relating the base values across transit systems with socio-economic and awareness indicators in each community. Four indicators are significant ($\alpha = 0.05$). We did not include Hispanic population ratio, median income, population density, and employment density in the table because of multicollinearity with the ratio of people with non-physical occupations. For the similar reason, we did not directly include the ratio of females due to multicollinearity with the ratio of African Americans. An F test shows the model is significant ($p < 0.001$). The R-squared value is 0.38 without

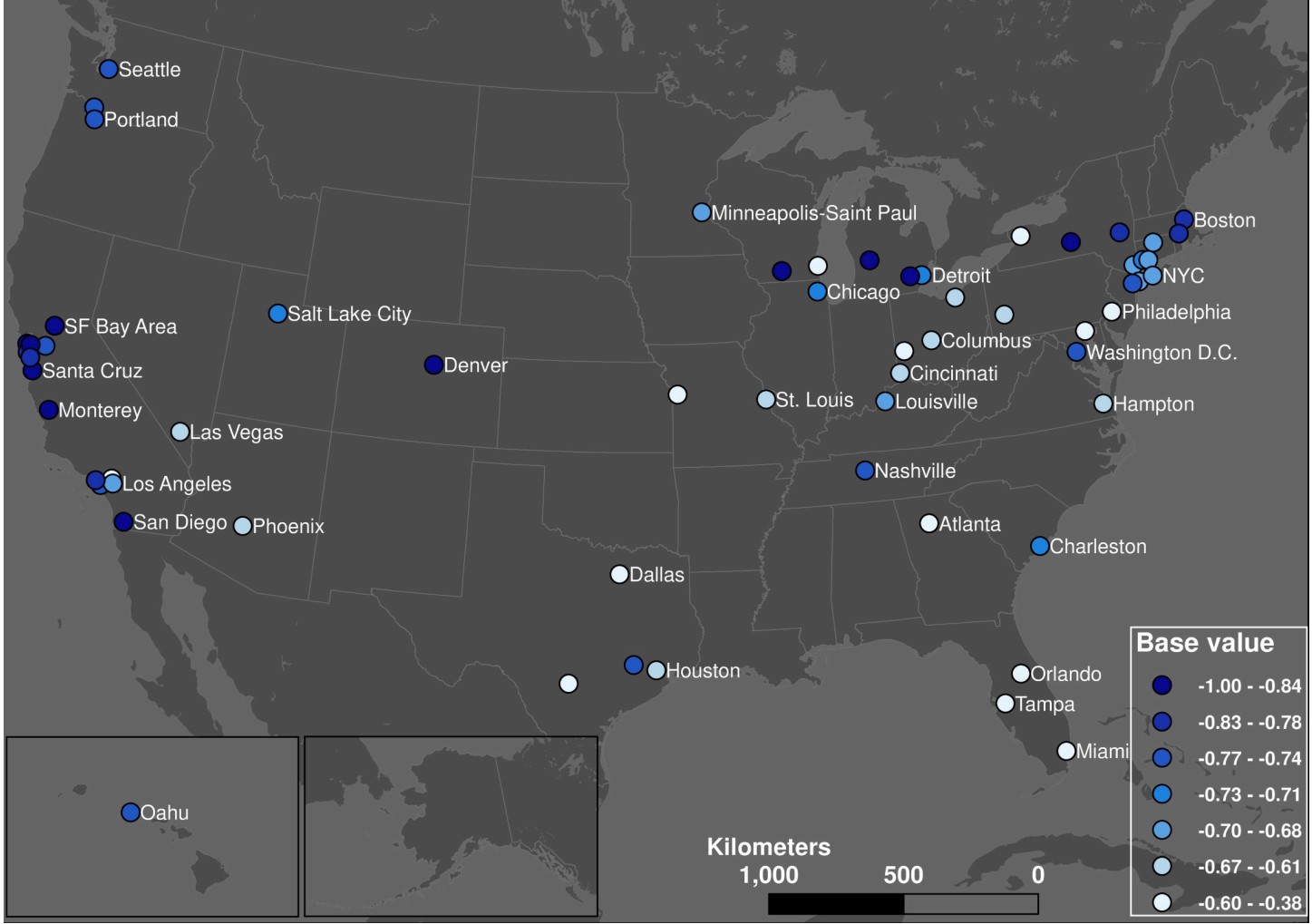

**Fig 4. The spatial distribution of base values across the United States (quantile classification).**

insignificant factors. A residuals assessment of the final model without the insignificant factors shows that the residuals are normally distributed and there are no lingering multicollinearity effects for significant factors. Further test also shows that the residuals of the model meet the homoscedasticity assumption. We also tried a spatial lag model: the results show that the

**Table 1. Results from regression analysis of base values with socio-economic and awareness indicators.**

|  | Coefficient | Standard error | t value | Pr(>\|t\|) | Variance Inflation Factor | R-squared increment |
|---|---|---|---|---|---|---|
| Intercept | -0.520 | 0.194 | -2.677 | 0.00861** | - | - |
| Ratio of people with non-physical occupation | -0.493 | 0.137 | -3.593 | 0.000497*** | 1.39 | 0.0728 |
| Ratio of African American population | 0.372 | 0.0812 | 4.576 | 1.29E-05*** | 1.26 | 0.118 |
| Ratio of population over 45 years old | 0.844 | 0.265 | 3.183 | 0.00192** | 1.08 | 0.0571 |
| Google search trend index | -0.00475 | 0.00238 | -2 | 0.0480* | 1.79 | 0.0225 |
| Ratio of people commuting to work | -0.240 | 0.197 | -1.215 | 0.227 | 13.31 | ~0 |
| Ratio of households with no vehicles | 0.307 | 0.177 | 1.732 | 0.0863 | 11.38 | 0.017 |

spatial autocorrelation parameter (rho) is not significant. These results indicate that it is appropriate to use ordinary least square regression model.

**Population with non-physical occupations.** The results confirm the hypothesis that greater decreases in transit demand are associated with a higher percentage of people with non-physical occupations. People who can work at home avoid public transit; people who cannot work at home and rely on public transit continue to use it.

Although we did not include the Hispanic population indicator due to multicollinearity, a significant negative correlation between ratio of Hispanic population and ratio of population with non-physical occupations suggests vulnerability of Hispanic populations during this health crisis: if a community has a higher Hispanic population, it is likely for the community to have a higher base value, which means more people use transit during the pandemic (presumably for work). This is also consistent with the occupation statistics: according to the labor force characteristics survey made by the US Bureau of Labor Statistics, the Hispanic population has the lowest percent (22%) of management, professional, and related occupations compared with White (41%), African American (31%), and Asian people (54%) in 2018 [37]. The correlation result also applies to the median income, population density, and employment density, which are all correlated with the ratio of population with non-physical occupations.

User surveys conducted by Transit app in April 2020 support these results. According to the survey, 92% of respondents who still use public transit report that they still use it to commute to work [28]. Meanwhile, the top-four occupation categories that are most likely to still using transit are production, installation, maintenance and repair, food preparation and service, and protective service. Although the categorizations of the survey and the ACS data are different, this is generally consistent with the occupation categories we derive from the ACS data.

The Transit survey also indicates that Spanish speakers are more likely to continue using the Transit app for trip planning purposes: English-language users dropped 71% from early February while Spanish-language users dropped by 50% over the same time period [29]. The income correlation is also confirmed by the survey results: compared with the survey results conducted by American Public Transportation Association (APTA) in 2017 [38], active users skew towards lower income brackets during the pandemic, especially for those whose annual income is less than $15000 [29]. The survey results provide first-hand evidence and reaffirm the correlation results about the vulnerability of Hispanic population and low-income population.

**Age.** The ratio of the population over 45 years old is associated with higher base values; older people in a community indicate higher levels of continued transit use during the pandemic. The Transit user survey also supports this result. By comparing the users' age composition in surveys conducted in September 2019 and April 2020, Transit found a drop in young people under 18 and between 25 to 44 years old; meanwhile, the relative ratio of people between 45 to 64 years old doubled [29].

**African American and female.** The regression analyses also indicates the dependence of African Americans on public transit, even during a pandemic. It is the most influential among the factors in Table 1 based on model fit (R-squared). There is a strong positive correlation between the ratio of African Americans in the population and the base value. These results are also consistent with the results of the Transit user survey. During the pandemic, African American people have the greatest share (>35%) of riders compared with other races in the US, while Caucasians were the majority (>40%) of the rider before the pandemic based on the 2017 APTA survey [28, 38]. The disproportionately small decrease of the African American population's transit demand supports the conclusion that cities with more African Americans are more likely to have a higher base value.

Higher base values are also highly correlated with larger ratios of females in the population; however, we also did not include it due to multicollinearity with the ratio of African Americans in the population. A higher ratio of females in the population is also correlated with lower income and a lower ratio of people with non-physical occupations. The Transit user survey supports these results in a dramatic manner. Among all the US users surveyed, the male and female proportions were roughly equal before the COVID-19 pandemic; during the pandemic, 56% are females while only 40% are males [28]. For some cities such as Philadelphia, more than 68% of riders are women. Meanwhile, Transit app users of color are also more likely to be females during the pandemic; more than 70% of the African-American riders during the pandemic are female [28].

**Awareness.** The Google search trend index for the keyword "Coronavirus" is significantly associated with the base value; cities with a higher search index tend to have a lower base value, as a higher search index means "Coronavirus" has a higher ratio among all Google searches in that region. This indicates the effects of people's awareness and concern on avoiding non-essential public transit trips. However, in Table 1, Google search trend index is not highly influential: it has the lowest R-squared improvement. This suggest that ethnicity, occupation, and age composition outweigh the awareness or preference when it comes to whether people will stay at home.

**Transit dependency.** The Transit user survey found 85% of users do not have access to a car [28], supporting that transit passengers during the pandemic are mostly captive passengers. Surprisingly, the ACS data on people using transit to commute and households with no vehicles do not have a significant correlation with base values. This suggests that the ACS dataset may not be a good metric for transit dependency during an emergency such as a pandemic. The presence of a vehicle in a household does not mean it is reliable, affordable to operate, or available to a given household member. Also, transit dependency is heterogeneous in many US cities: while most residents are not transit dependent, there are neighborhoods with concentrated poverty and transit dependence [39, 40]. The user survey shows that access to private vehicles is highly heterogeneous for different household income levels [28].

## Response intervals

Fig 5 shows the distribution of the response interval measures in the US relative to the cliff and base time points with no incubation lag. For cliff point response intervals, the pattern is highly polarized. In some cities with international airports, such as Seattle where the first US COVID-19 cases were found, people still used transit after the first case emerged. Meanwhile in other cities, such as most cities in the Midwest with the exception of Chicago, people started avoiding transit trips in advance of confirmed local community spread. This may be due to Seattle's precedence in COVID-19 spread in America: the media began to report the severity of this disease and the CDC made the prediction that the community spread is inevitable near the end of February 2020 [41].

The response interval patterns in Fig 5 suggest that initial declines in public transit usage may have limited the spread of the disease in some communities; 58% of all transit systems have a positive response interval. However, the picture is less sanguine after we factor in incubation lag: the ratio of positive response interval decreases to 32%. New York City is an illustrative example. With a lag of zero days, six of 13 systems in the New York City area have positive response intervals, suggesting declines in transit usage in advance of community spread. With a lag of five days, all 13 transit systems have negative response intervals, meaning the virus could have been spreading in the community before any appreciable decline in transit demand. In contrast, most transit systems in the Midwest such as Missouri, Ohio, Michigan,

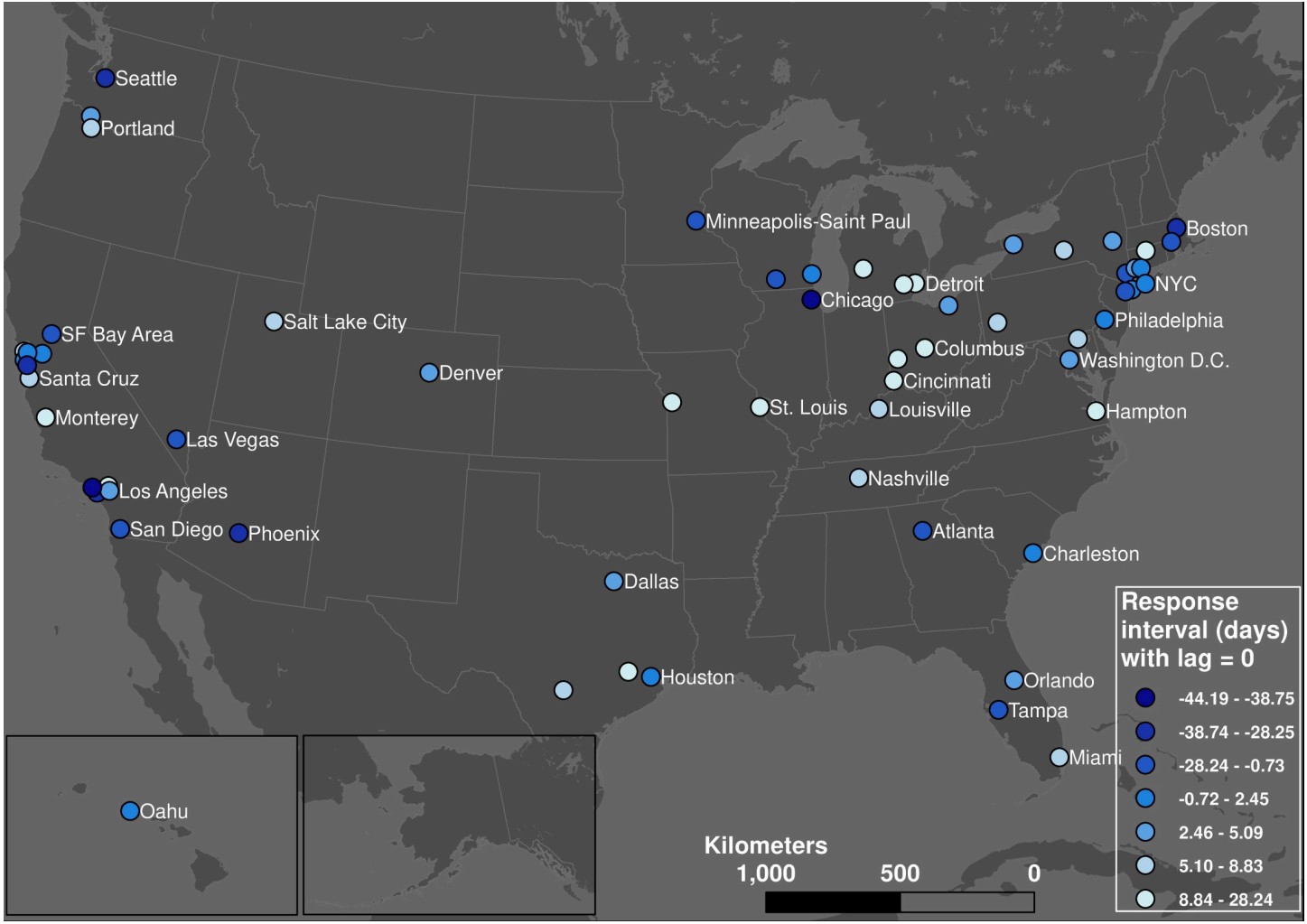

**Fig 5. The geographic pattern of response interval with incubation lag = 0 from cliff point (quantile classification).**

and Kentucky still have positive response intervals with an incubation lag of five days. This is supported by cellphone location data: those Midwest states above had known stay-at-home orders before March 27[th] and the measured trips are significantly less [42]. For a more conservative scenario of an incubation lag = 14 days, most transit systems (93.8%) have negative response intervals.

However, the geographic pattern is highly homogeneous for the response intervals from base point, which represent how early each transit system's users finished the stay-at-home process compared to the community spread. With an incubation lag of 0 days, only Capital Metro in Austin, Texas has a positive response interval. For the case of Austin, the demand decrease started on March 5[th] and finished by March 23[nd]; the first case was confirmed on March 25[th]. However, long before the first confirmed case, the city and county authority declared a local state of emergency on March 6[th] [43], which is one of the earliest places to take actions in the United States. The cliff point is also the same as the date of the local state of emergency, which suggests the effectiveness of the executive order. This can be one reason for the relatively fast and earlier reaction of transit users. The correlation between the cliff point and the date of declaring state emergency moreover support this claim: it shows that an earlier

cliff point is positively correlated (p = 0.002) with an earlier emergency date. It suggests that transit users' response times are not synchronized with the evolution of the pandemic, but with the local community's actions and policies.

### Decay rate

Fig 6 shows the geographic pattern of decay rate. Transit systems in the north, especially those in larger communities, and college towns reached their base values the quickest while transit systems in the Midwest and southern communities took the longest to reach their bases. College towns emptied quickly during the pandemic. The slower decay rate in the Midwest and South may be explained by businesses staying open longer during the pandemic.

Fig 7 shows the decay rate has a positive hyperbolic correlation with cliff point and a negative hyperbolic correlation with base point, as formulas (5) and (6) suggested. This indicates that the later the demand decline happened, the faster it occurred, and the decline process finished earlier. This could be because the general transit passengers may be more aware of the risk of COVID-19 when more cases are reported nationally; the perceived fear grows higher as time passes, causing transit users to act faster and reach the base point earlier. This also

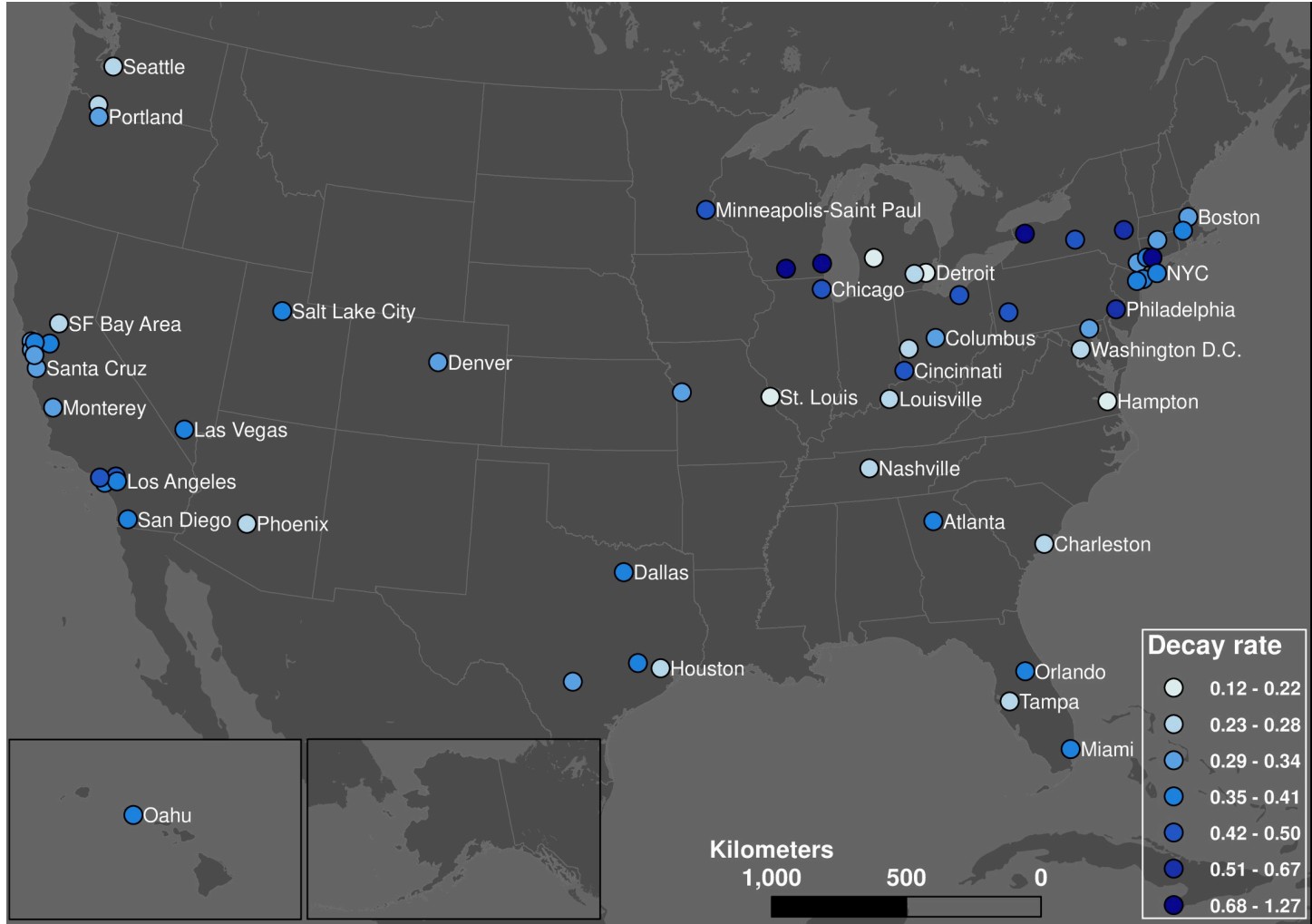

**Fig 6. Geographic distribution of decay rate (quantile classification).**

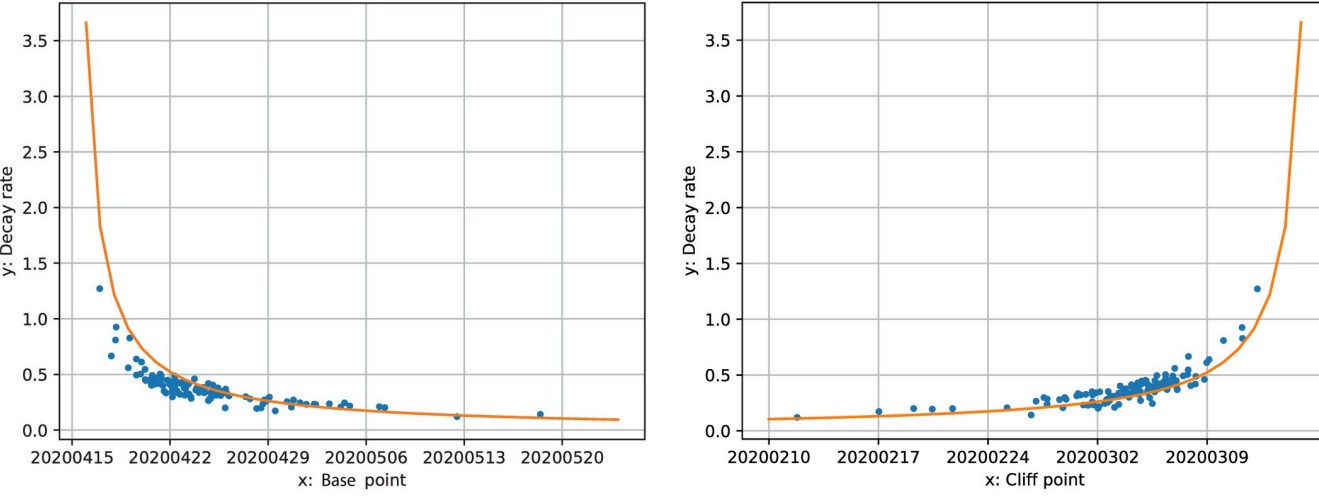

**Fig 7. Hyperbolic relationship between decay rates and cliff/base points.**

suggests that the time when the decline finished is less related to when it started than to the speed of reaction. The major determinant of the cliff and base point in formula (5) is decay rate instead of $t_0$.

## Change in daily transit demand dynamics

Overall, we observe a relationship between a change in hourly demand patterns (measured by the Procrustes distance) and the total drop in demand due to COVID-19 (measured by the base value). Fig 8 shows the geographic distribution of each transit system's average Procrustes distance between its normal and pandemic hourly demand curves. This map shows a similar pattern to the geographic distribution of base values (Fig 4): transit systems serving communities that are dominated by non-physical occupations (including university towns) experienced large qualitative changes in their weekday hourly demand patterns. In contrast, the Procrustes distances between normal and pandemic hourly transit demand profiles of communities in the Midwest and Northeast is low, meaning these transit systems retained much of their typical daily demand profile (albeit with lower levels of overall demand). Fig 9 confirms the strong correlation between the Procrustes distance and base values: higher levels of base demand during the pandemic also means less shift from the typical hourly demand profile.

Fig 10 shows the daily distribution of all the transit systems' average Procrustes distance between its normal and pandemic hourly demand curves. We see a pattern of a period of increasing difference during first few weeks, re-stabilization at a higher level, and a signal of decline at the very end. This means that the hourly demand dynamics gradually diverge from the normality, stabilize, and then show signs of returning to normal.

The Procrustes distance value also shows a regular periodical pattern: the distances are higher for weekdays (black points in Fig 10) than weekends (blue points in Fig 10), which means the hourly demand pattern diverged from normal more on weekdays than weekends. By visualizing the hourly demand pattern, we note that weekday and weekend hourly demand patterns became more similar. To confirm this, we calculate the Procrustes distances between weekdays and weekends. These distances decreased for all transit systems during the pandemic. Two factors could explain this convergence between weekday and weekend hourly demand patterns. First is the disproportional decrease of the morning and afternoon commuting activities in the weekdays. This change will generally flatten the peaks and diminish the

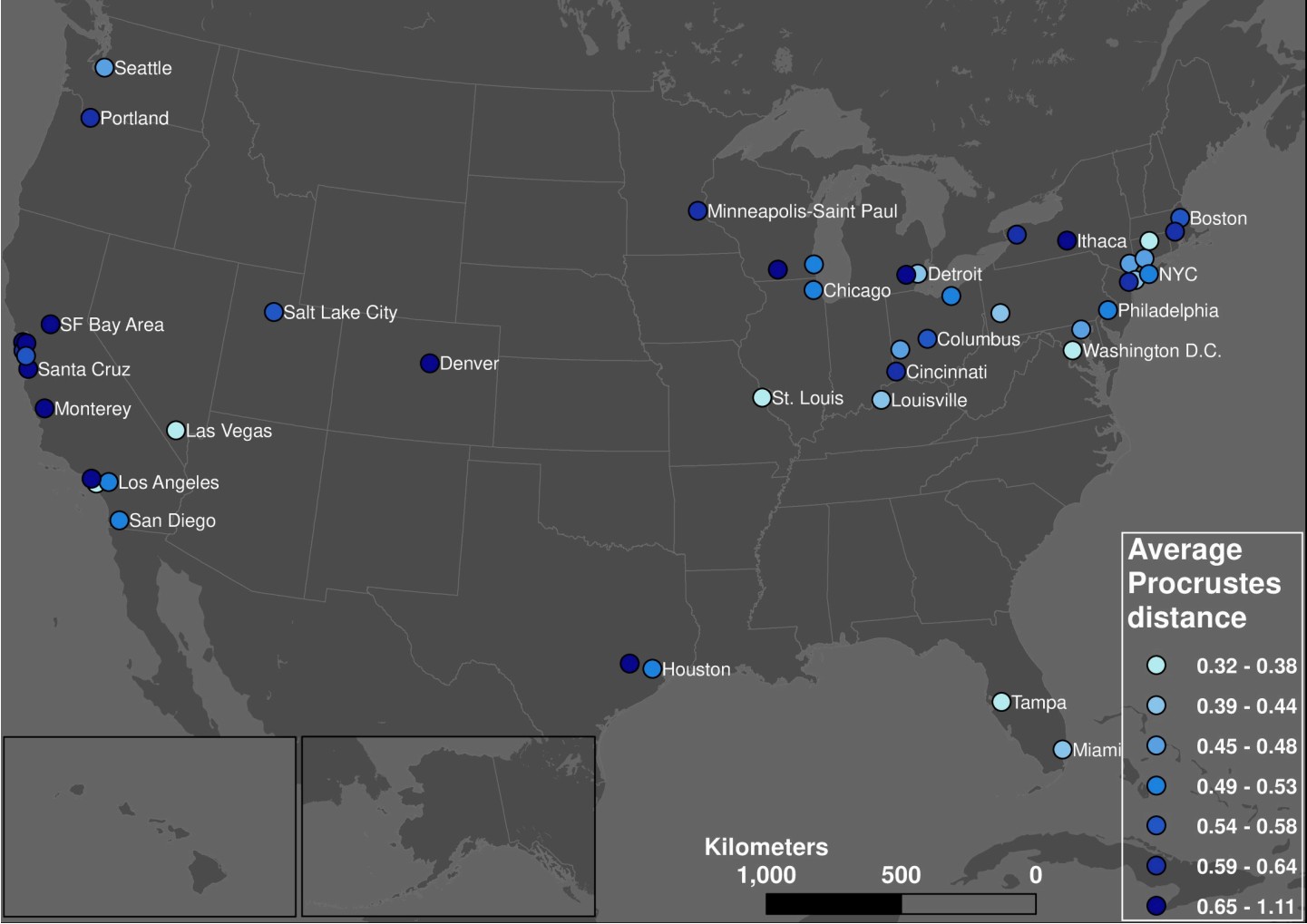

**Fig 8. Geographic distribution average Procrustes distance between normal and pandemic weekday hourly demand curves.**

contrast between normal hours and rush hours. This is driven by the privileged population with non-physical occupations: their absence made weekdays more like weekends. Second, the reduction of non-essential activities, such as leisure and shopping trips, also make commuting-relevant trips more prominent during the weekends. This effect is especially obvious in the New York City: its population highly relies on public transit and the non-physical occupation rate is not high. For example, for the Metropolitan Transportation Authority (MTA) systems, the curves of Sundays usually have one peak during 2–4 pm; however, the shape of the Sunday curves during the pandemic had two peaks, which was similar to the weekdays' commuting pattern. This process is also driven by transit dependent essential workers with jobs that do not correspond to traditional 9am-5pm weekday schedules. These two factors homogenized each day of week and make the boundary between weekends and weekdays less obvious.

## Discussion

The minimum value of the transit demand curve, *base value*, is an indicator of transit as an essential service: it shows continued use transit system regardless of the pandemic; most likely

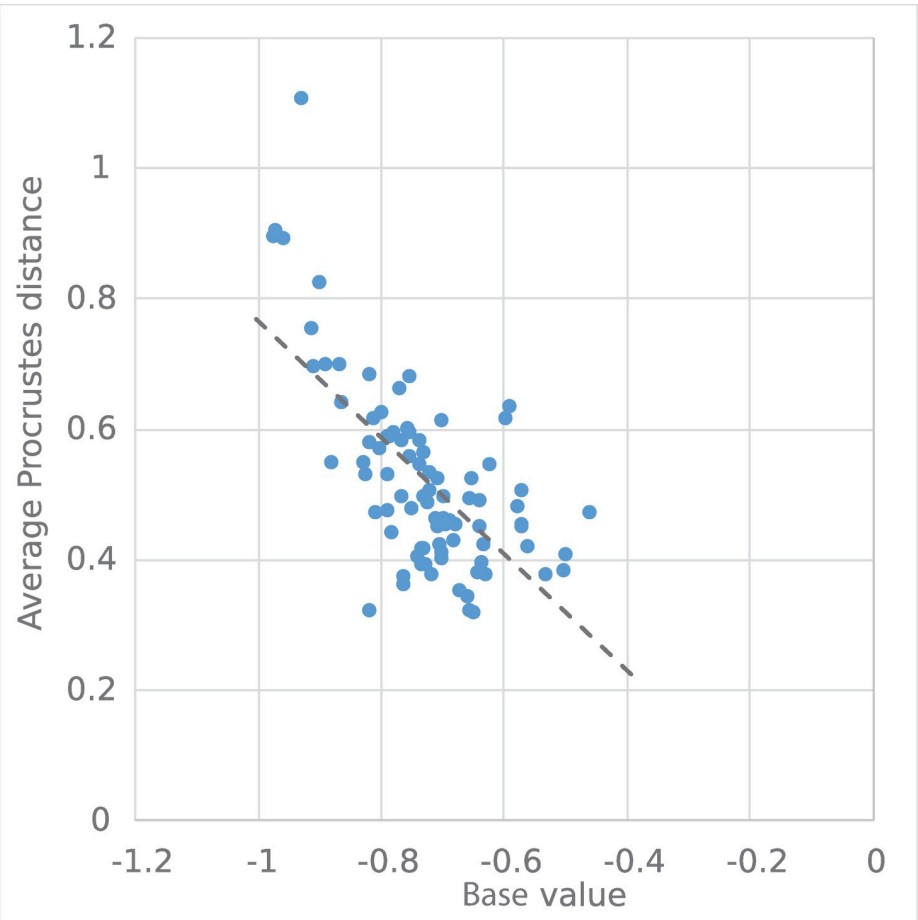

**Fig 9. Relationship between average Procrustes distance and base value.**

people who are transit dependent and perform essential jobs and other activities. Geographic patterns shows communities dominated by the information sector and universities have lower base values. Communities with less non-physical occupations ratio, larger ratio of minority population (African American, Female, Hispanic), more middle-age and senior people over 45 years old, and fewer "Coronavirus" searches tend to have higher base value. The user demographic survey conducted by Transit app supports these conclusions. These results affirm a stark fact: cities with more essential workers and a more vulnerable population tend to maintain higher transit demand levels during COVID-19. This moreover suggests the necessity of the transit system during a pandemic when transit systems lose a great deal of discretionary demand. This should motivate transit planners, policy makers, political leaders, and taxpayers to rethink the role of transit systems not as a business, but as critical infrastructure for a community.

It is noteworthy that base values are not associated with the ratio of transit commuters and households with no access to private vehicles from the US Census American Community Survey. This suggests that these commonly used measures are not adequate for describing essential transit demand during a crisis such as a pandemic. These variables may not capture transit dependence since transit commuters during normal times include both choice and dependent riders. Also, having at least one vehicle per household does not mean that individuals have ready access to reliable transportation. This suggests a need for developing more accurate measure of transit dependency for use in crises.

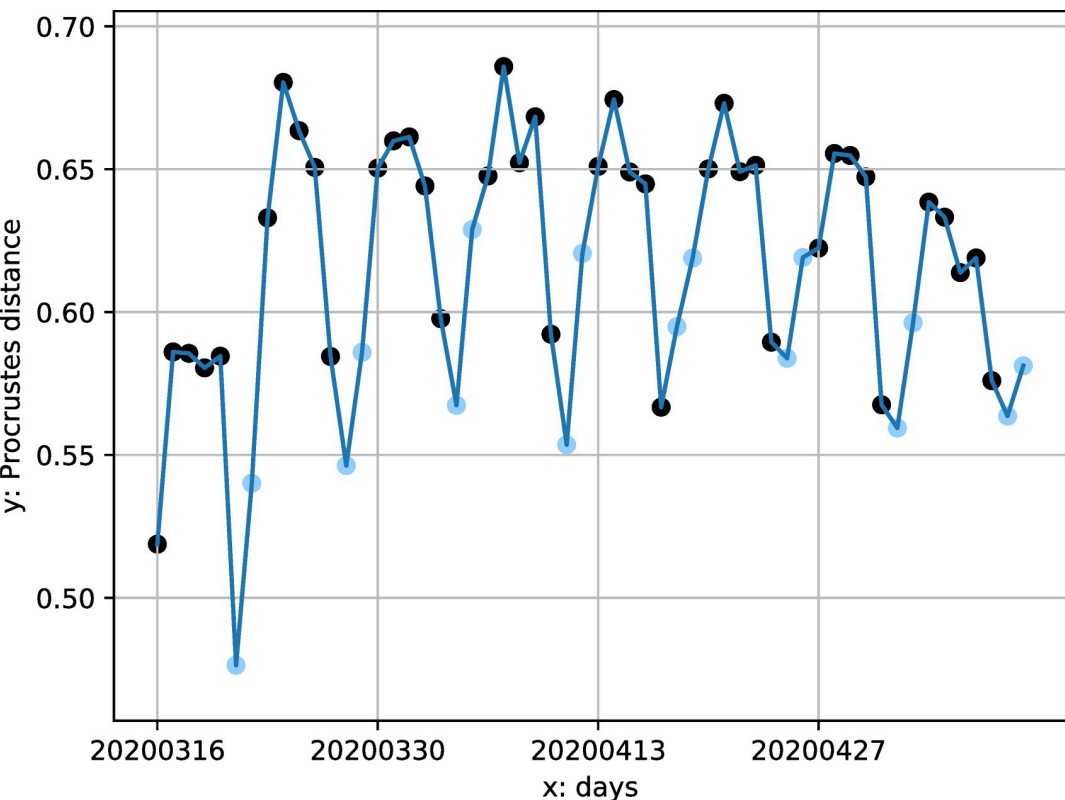

**Fig 10. Temporal distribution of all transit system's average Procrustes distance between normal and pandemic curves.**
Black points: weekdays; blue points: weekends.

The start and end date of transit demand curve, *cliff point* and *base point*, show when discretionary transit demand—people who can work at home or do not work, and people who have other mobility options—started and finished its decline due to COVID-19. Results suggest that that response time is not synchronized with the progression of the disease in a community; instead, the cliff point is correlated with the date of declaring state of emergency for each state. 58% of all transit systems experienced a decline in discretionary travel demand before local community spread started. However, this pattern is less sanguine when we factor in incubation lags: the number drops to 32% with the reported median incubation period of five days and moreover drops to 6% with the reported maximum incubation period of 14 days. Meanwhile, almost no transit systems' discretionary demand cleared fast enough to reach the base point before community spread.

The *decay rate* shows the speed of the decline process. Our results suggest that faster decay rate associates with later cliff point and earlier base point. This may be due to growing awareness as time passed, causing people to act faster. This also suggests that the end date of the decline (base point) is mostly determined by the reaction speed (decay rate) but not the start date of the decline (cliff point).

The commuting analyses based on hourly transit demand data show that essential passengers' commuting routine during the pandemic is substantially different from the normal routine. Differences between the normal and pandemic weekday demand profiles is geographically polarized and highly correlated with the base value: systems with higher base values (e.g., communities in the Northeast and Midwest) retain more of their hourly demand pattern than systems with lower base values (e.g., communities with a large number of non-

physical occupations, including cities in California, and university towns where a large proportion of the population left). The impact on hourly demand profiles increased as the pandemic developed. The pandemic also made the weekdays and weekends more similar to disproportional decrease of the morning and afternoon commuting activity, making the difference between rush hours and normal hours less obvious; meanwhile weekdays became more like weekends because the cessation of non-essential businesses made the weekends trips more dominated by commuting.

## Conclusion

This paper is a first approximation for understanding the differential impacts of a major pandemic such as COVID-19 on public transit systems in the United States. We use activity data from a widely used public transit navigation app to measure and compare changes in demand for 113 transit systems across the United States during the COVID-19 pandemic. We fit logistic curves describing declines in daily transit system demand and derive parameters describing the decline dynamics. We also compare differences in hourly demand profiles on weekdays and weekends. We find differential impacts across communities and social groups, indicating differences in public transit dependency that can be explained by social factors in each community. Our study highlights public transit as a critical infrastructure for a community during a pandemic and the vulnerabilities of some underprivileged social groups (women, Hispanic, African-Americans) as they travel to perform essential activities.

Additional research should build on this study to resolve some of its limitations and more deeply investigate the patterns discovered. One limitation of our study concerns the representativeness of the transit demand data for actual ridership. We use data from the Transit app as a surrogate for demand since actual passenger counts for systems at a national level are difficult to obtain. Although a test between official ridership data and the transit demand data for some systems suggest no significant differences overall, transit system-level comparisons should nevertheless be viewed as tentative. Data from automated passenger counting technologies, smart card or other transit pass data would allow more definitive comparison, albeit at a system level and not at the national level as in this paper.

Another limitation concerns the geographic resolution of the data is each transit system. We make our comparison using system level data and its corresponding county-equivalent. This can mask important differences within each system (e.g., route-by-route changes) and across neighborhoods within each community. Again, this calls for a deeper investigation within each system.

Finally, there is an urgent need for attitudinal and behavioral surveys and further analysis to confirm some of the patterns suggested in this study about ridership during a pandemic, individuals' perceptions and their reactions. With this more nuanced understanding of individual public transit behavior during a pandemic, we can help design effective public transit systems that meet the needs of vulnerable passengers using transit to perform essential activities, creating transportation systems that are more inclusive and resilient to shocks.

## Author Contributions

**Conceptualization:** Luyu Liu, Harvey J. Miller.

**Data curation:** Luyu Liu, Jonathan Scheff.

**Formal analysis:** Luyu Liu.

**Funding acquisition:** Harvey J. Miller.

**Investigation:** Luyu Liu, Harvey J. Miller.

**Methodology:** Luyu Liu.

**Project administration:** Luyu Liu.

**Resources:** Luyu Liu, Harvey J. Miller.

**Software:** Luyu Liu.

**Supervision:** Harvey J. Miller.

**Validation:** Luyu Liu.

**Visualization:** Luyu Liu.

**Writing – original draft:** Luyu Liu.

**Writing – review & editing:** Luyu Liu, Harvey J. Miller, Jonathan Scheff.

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
