## [Decision Letter · Decision Letter 0]

20 Jul 2020

PONE-D-20-16721

The Impacts of COVID-19 Pandemic on Public Transit Demand in the United States: An Analysis Using Smartphone Data

PLOS ONE

Dear Dr. Liu,

Thank you for submitting your manuscript to PLOS ONE. After careful consideration, we feel that it has merit but does not fully meet PLOS ONE’s publication criteria as it currently stands. Therefore, we invite you to submit a revised version of the manuscript that addresses the points raised during the review process.

Please properly address the two reviewers' comments are attached. In addition, please also note the innovative aspects of the research and the spatiotemporal dimension in the revised version. 

We look forward to receiving your revised manuscript.

Kind regards,

Chaowei Yang

Academic Editor

PLOS ONE

Journal Requirements:

2. In ethics statement in the manuscript and in the online submission form, please confirm that only publicly available data have been used in your study, and that no users' personal information have been accessed or collected.

4.  Thank you for stating the following in the Financial Disclosure section:

We note that one or more of the authors are employed by a commercial company: "Transit App Inc.,"

5. Please amend either the title on the online submission form (via Edit Submission) or the title in the manuscript so that they are identical.

6. Please remove your figures from within your manuscript file, leaving only the individual TIFF/EPS image files, uploaded separately.  These will be automatically included in the reviewers’ PDF.

7. Please ensure that you refer to Figure 4 in your text as, if accepted, production will need this reference to link the reader to the figure

8. We note you have included a table to which you do not refer in the text of your manuscript. Please ensure that you refer to Table 1 in your text; if accepted, production will need this reference to link the reader to the Table.

Reviewers' comments:

Reviewer's Responses to Questions

**Comments to the Author**

1. Is the manuscript technically sound, and do the data support the conclusions?

Reviewer #1: Yes

Reviewer #2: Yes

2. Has the statistical analysis been performed appropriately and rigorously? 

Reviewer #1: No

Reviewer #2: Yes

3. Have the authors made all data underlying the findings in their manuscript fully available?

Reviewer #1: No

Reviewer #2: Yes

4. Is the manuscript presented in an intelligible fashion and written in standard English?

Reviewer #1: Yes

Reviewer #2: No

5. Review Comments to the Author

Reviewer #1: This paper studied an interesting and timely research question regarding the transit demand change during the COVID-19 pandemic. The authors employed the data from Transit App to capture transit demand and derived various indexes to describe the change patterns. Overall, this study offers timely data analytics to monitor transit demand during COVID-19. However, there are still several notable concerns with this paper. Detailed comments follow:

The methodological contribution of this paper is limited. Most analyses conducted in this study are descriptive, and the whole paper lacks convincing and strict model build and description:

1) The authors employed a logistic function to fit the transit demand data for each transit system. First, the logistic function is quite different from the logistic model, the authors should be careful when describing their methods. Second, I failed to found any results of the logistic functions. The authors should at least give a summary of the fitting accuracy and statistical significance of the logistic functions for different transit systems.

2) When modeling the factors related to floor value, some essential variables are missing. For example, the population density, the job density, and the factors related to transit accessibility (for example, the number of transit stations in each city. The data can be derived from OSM POI). The authors should do more literature review regarding the built environment and public transit to understand which covariates are essential.

3) Is the simple linear model appropriate to fit the floor value? Do the data meet the normality assumption? How to handle spatial auto-correlations? The authors should address these issues before using an OLS model.

4) Why the authors only build a model for floor value, while ignoring the other indexes like cliff and floor points, response intervals, the decay rate?

The visualization part is insufficient also. At least two figures are important but missing. First, a figure of the transit demand varying patterns across the study period. Second, a figure visualizing the observed data versus the fitted data using the logistic function. The indexes like floor value, cliff and floor points, response intervals, the decay rate, can also be annotated in the figures.

Some other minor comments:

1) The authors should involve a proofreader to improve writing. Many words are unprofessional and hard to understand. For example, the floor value mostly means the closest integer less than or equal to a given number, rather than the lowest plateau value the authors want to express.

2) The holidays should be excluded from the study periods due to the unusual human mobility patterns.

3) The authors should also report the variables with insignificant P-values in Table 1.

4) In Line 353, why does the ratio of female have high multi-collinearity with the ratio of African Americans?

Reviewer #2: This interesting paper investigates the impacts of the COVID-19 pandemic of public transit ridership across major systems in the US. The main data supporting this research are provided by the Transit app. It is a very timely effort, focusing on an important topic with a strong tie to the society. Methods are adequate. But I have several concerns for the authors to address.

1. Introduction section is not very motivating. For example, why is it important to study the changes in public transit ridership, along with some metrics like the floor value and so on? What knowledge do we gain from this? How can this knowledge be beneficial to the society? The authors could’ve done a better job discussing these points.

2. My major concern is that there is no lit review in this paper. Without discussing previous studies of relevant scopes, how can we know the research gap and the contributions of this work? It is important to add such a section to back up your ideas.

3. Variables. The authors should justify why some variables are selected. I am concerned about a few varaibles. One such variable, for example, is the occupation type factor. As described in lines 168-169, “Information, Financial activities, and professional and business service” were selected and adopted in the model. The assumption, as detailed in lines 164-165 and line 169, is that these types of workers are more likely to work from home during this pandemic and thus areas with more of these workers are more likely to experience a greater hit in ridership. This assumption/assertion is somehow problematic. I think these subgroups are less likely to use public transit but instead rely more on private vehicles before this pandemic. That said, they may not be an important component to the typical ridership. Therefore, looking at communities with higher percentage of these workers for examining sudden ridership change is less convincing.

In addition to the variables already included in the model, I think the number of homeless people should be considered. Homeless people are more likely to take/occupy public transit, especially in large cities like NYC. As this particular subgroup of population reportedly has higher infection risk, the related transit systems may be affected more severely. This can also be related to the awareness factor discussed in the paper.

4. Provide more details. Throughout the paper, the authors claimed that the Transit app is a widely used app. The only statements related to this is in lines 99-101—“the app covers over 200 cities aournd the world with … download on…” This is insufficient to back up the point that it is a widely used app, and thus leading me to question the representativeness of the data. As the study area is the US, so the authors should provide more details about the user coverage and usage stats (ideally some comparisons with other competitors for showing its market share) to define how “widely” it is being used in the US.

More details about methods/analyses. Section 2 describes the analyses/methods, but I find it a bit loosely connected. More details should be provided to better connect these steps and help readers get the full picture.

I think it would be great if every city in the maps is labeled.

5. Figure 1. Why COVID curve (orange) is more prominent than the typical curve (blue)?

6. There are many typos and formatting issues in the paper, making it difficult to read. The language should be improved.

6. PLOS authors have the option to publish the peer review history of their article (what does this mean?). If published, this will include your full peer review and any attached files.

Reviewer #1: **Yes: **Chenfeng Xiong

Reviewer #2: No

---

## [Author Response · Author response to Decision Letter 0]

11 Aug 2020

Please find our responses to the reviewers in the attached files.

---

## [Decision Letter · Decision Letter 1]

4 Nov 2020

The impacts of COVID-19 pandemic on public transit demand in the United States

PONE-D-20-16721R1

Dear Dr. Liu,

We’re pleased to inform you that your manuscript has been judged scientifically suitable for publication and will be formally accepted for publication once it meets all outstanding technical requirements.

Kind regards,

Chaowei Yang

Academic Editor

PLOS ONE

Additional Editor Comments (optional):

Reviewers' comments:

Reviewer's Responses to Questions

**Comments to the Author**

1. If the authors have adequately addressed your comments raised in a previous round of review and you feel that this manuscript is now acceptable for publication, you may indicate that here to bypass the “Comments to the Author” section, enter your conflict of interest statement in the “Confidential to Editor” section, and submit your "Accept" recommendation.

Reviewer #2: All comments have been addressed

Reviewer #3: All comments have been addressed

2. Is the manuscript technically sound, and do the data support the conclusions?

Reviewer #2: (No Response)

Reviewer #3: Yes

3. Has the statistical analysis been performed appropriately and rigorously? 

Reviewer #2: (No Response)

Reviewer #3: Yes

4. Have the authors made all data underlying the findings in their manuscript fully available?

Reviewer #2: (No Response)

Reviewer #3: Yes

5. Is the manuscript presented in an intelligible fashion and written in standard English?

Reviewer #2: (No Response)

Reviewer #3: Yes

6. Review Comments to the Author

Reviewer #2: I am happy with the revision. One minor suggestion though. To map all the cities with labels in one map, I guess the authors could make several inset maps zooming into particular regions such as Los Angeles so that labels do not clutter.

Reviewer #3: This manuscript represented a timely effort that show public transit demand change in the U.S. during the pandemic with Transit app dataset by adopting logistic statistical analysis. All comments from first round reviewing have been addressed propriety. This revised manuscript has met the standard for further proceeding.

7. PLOS authors have the option to publish the peer review history of their article (what does this mean?). If published, this will include your full peer review and any attached files.

Reviewer #2: **Yes: **Yujie Hu

Reviewer #3: No

---

## [Editor Report · Acceptance letter]

9 Nov 2020

PONE-D-20-16721R1 

The impacts of COVID-19 pandemic on public transit demand in the United States 

Dear Dr. Liu:

I'm pleased to inform you that your manuscript has been deemed suitable for publication in PLOS ONE. Congratulations! Your manuscript is now with our production department. 

Kind regards, 

on behalf of

Dr. Chaowei Yang 

Academic Editor

PLOS ONE